# Signage Interventions for Stair Climbing at Work: More than 700,000 Reasons for Caution

**DOI:** 10.3390/ijerph16193782

**Published:** 2019-10-08

**Authors:** Anna Puig-Ribera, Anna M. Señé-Mir, Guy A. H. Taylor-Covill, Núria De Lara, Douglas Carroll, Amanda Daley, Roger Holder, Erica Thomas, Raimon Milà, Frank F. Eves

**Affiliations:** 1Departament de Ciències de l’Activitat Física, Centre d’Estudis Sanitaris i Socials, Universitat de Vic-Universitat Central de Catalunya, 08500 Barcelona, Vic, Spain; 2School of Sport, Exercise and Rehabilitation Sciences, University of Birmingham, Birmingham B15 2TT, UK; 3Agència de Salut Pública de Catalunya, 08023 Barcelona, Spain; 4Department of General Practice, University of Birmingham, Birmingham B15 2TT, UK; 5Departament de Salut i Acció Social, Universitat de Vic-Universitat Central de Catalunya, 08500 Barcelona, Vic, Spain

**Keywords:** stair climbing, stair descent, point-of-choice prompts, workplace, pedestrian movement, lifestyle physical activity

## Abstract

Increased stair climbing reduces cardiovascular disease risk. While signage interventions for workplace stair climbing offer a low-cost tool to improve population health, inconsistent effects of intervention occur. Pedestrian movement within the built environment has major effects on stair use, independent of any health initiative. This paper used pooled data from UK and Spanish workplaces to test the effects of signage interventions when pedestrian movement was controlled for in analyses. Automated counters measured stair and elevator usage at the ground floor throughout the working day. Signage interventions employed previously successful campaigns. In the UK, minute-by-minute stair/elevator choices measured effects of momentary pedestrian traffic at the choice-point (*n* = 426,605). In Spain, aggregated pedestrian traffic every 30 min measured effects for ‘busyness’ of the building (*n* = 293,300). Intervention effects on stair descent (3 of 4 analyses) were more frequent than effects on stair climbing, the behavior with proven health benefits (1 of 4 analyses). Any intervention effects were of small magnitude relative to the influence of pedestrian movement. Failure to control for pedestrian movement compromises any estimate for signage effectiveness. These pooled data provide limited evidence that signage interventions for stair climbing at work will enhance population health.

## 1. Introduction

### 1.1. Background

Physical inactivity is a major risk factor for cardio-metabolic disorders and some cancers [1]. Accumulating physical activity as part of daily life is one promising approach to increase population levels of activity [2,3,4]. In addition to individual factors, the potential effects of the built environment on lifestyle activity is a current topic of considerable interest [2,4]. The broad aim of this approach is to identify environments that would encourage more active communities. One frequently researched lifestyle activity is stair usage. Stairs are regularly encountered in the built environment, particularly at work, and choosing them rather than escalators or elevators replaces inactivity with a bout of activity. In the seminal Harvard Alumni studies, stair climbing was linked prospectively to reduced risk of heart attacks and stroke [5,6], with subsequent experimental increases in climbing improving the cardiovascular disease risk factors of blood pressure, lipoproteins, abdominal fat, and aerobic fitness [7,8,9,10]. Increased stair climbing benefits population health and is promoted by public health agencies in the developed world [11,12]. At work, stairs are almost ubiquitous. Increased use to climb would improve employee’s health at minimal cost to the parent organization.

To increase stair climbing, typically a health promotion message is installed at the point-of-choice between stairs and the mechanized alternative. These simple, low-cost interventions change the environment in which behavior occurs by highlighting a potential health benefit [13]. The message attempts to disrupt habitual choice of the escalator or elevator at a time and place where a more active choice can be made. Reviews consistently conclude that these prompts can increase stair usage in public access settings (e.g., stations) and also at work [3,14,15,16,17]. The recent Physical Activity Guidelines Advisory Committee report concluded that strong evidence demonstrated ‘prompts to use stairs versus escalators or elevators are effective’ [17] (p. 67). Nonetheless, the environmental context is important [18,19]; typically, elevators are found in workplaces, whereas escalators are more common in environments such as stations. Choice of the stairs rather than the escalator in a station is not the same choice for public health as choosing stairs at work. A commuter responding to prompts in a station adds only two extra climbs a day [18,19]. In contrast, an employee at work prompted to use the stairs could accumulate repeated climbing episodes throughout the day. Workplace climbing could provide appreciable physical activity towards a public health dividend from these simple interventions [19]. As a result, the effectiveness of signage interventions at work is key evidence for public health dividends from these environmental prompts.

To test effectiveness, prompting interventions use quasi-experimental designs, with pre/post changes as evidence. For logistical reasons, there are no randomized control trials and only one study with a control group for signage [20]. Generalized estimates of effectiveness are potentially problematic without control comparisons. For quasi-experimental designs, any uncontrolled influences on the measured outcome could provide rival, alternative explanations for change. Pedestrian movement within the built environment influences stair choice independently of any intervention (e.g., [13]). This movement is not controllable.

### 1.2. Estimates of the Effectiveness of Signage Interventions at Work

A recent sequential meta-analysis estimated effects on stair climbing of signage at +2.2% when combining results from public access and workplace settings [14]. Heterogeneity between studies was high, *I*^2^ = 97.5%. When comparing the two settings, the most recent estimate by Bellicha and co-workers used medians (i.e., the study at the midpoint when the effects in different interventions were ordered from low to high) [3]. The median is a plausible alternative measure of effectiveness across the diverse interventions and settings that produce heterogeneity. Bellicha and colleagues reported a ‘median absolute increase in stair use of approximately +4%’ in both public access and workplace settings [3] (p. 6). Nonetheless, many large sample studies (i.e., those with greater statistical power) reported no or equivocal evidence for increased stair usage [21,22,23,24,25,26,27,28]. Median estimates of effectiveness discount these larger sample studies. Further, any estimate based on the median could be inflated by missing studies from the less effective tail of the distribution due to publication bias or residence in a file draw. A sample-size weighted average of the magnitude of change is an estimate of effect that appropriately incorporates sample size, unlike medians [18]. The sample-size weighted average increase for the stair climbing data at work in the review was +1.6% (total *n* = 379,491), considerably lower than the median of +4.3% estimated by Bellicha and co-workers [3]. Given this low magnitude, any uncontrolled effects on stair choice are important to any estimate of effectiveness.

### 1.3. Effects of Pedestrian Movement on Stair Usage

Movement of pedestrians using a building will influence stair behavior, independent of any signage intervention [13,29]. One variable with ubiquitous effects on stair climbing is the volume of pedestrian traffic at the choice-point. For choice between escalators and stairs in public access settings, increased traffic volume at the choice-point increases stair climbing [13,30,31]. Pedestrians are less likely to wait for a busy escalator in stations and shopping malls, and the stairs provide a quicker method of ascent [30]. These effects of pedestrian traffic volume are of large magnitude relative to the effects of signage [32]. Effects of pedestrian movement offer a rival, alternative explanation for changes in stair climbing in public access settings that has been controlled for repeatedly in previous analyses (e.g., [13,31,32]).

In workplaces, where the alternative is an elevator, the opposite effect of pedestrian movement occurs. Increased momentary traffic at the elevator choice-point (i.e., each minute) reduces stair climbing [26,29,33]. We suspect this reduction reflects effects of time pressure and potential social interactions at the choice-point [18,26,29,33,34,35]. The need to wait for an elevator that might be elsewhere in the building means that the choice at work is not the same as when the alternative is an adjacent escalator in public access settings [18,29,34]. Two further variables related to pedestrian movement also affect stair climbing. The more people in the building, the more likely the occupants will choose the stairs [26,29]. When a building is busy, waiting times for the elevator increase, and stairs may represent the quicker route to the destination [29]. Finally, employees ascend into the building in the morning and descend out of it in the afternoon [29]. Across the day, stair climbing reduces, but stair descent has been reported as both increasing and reducing [26,29]. These uncontrolled effects of pedestrian movement in the workplace offer rival alternative explanations for any changes in stair usage. This paper used pooled data to estimate effects of signage at work when pedestrian movement throughout the day was controlled for in analyses for the first time.

### 1.4. Stair Climbing or Stair Usage

The major question for public health agencies is the efficacy of signage interventions on cardiovascular disease risk outcomes. Some studies combine stair ascent and descent into a measure of stair usage when estimating the effect of intervention [3]. For efficacy, however, it is only stair climbing that has been linked to reduced cardiovascular disease risk [5,6,7,8,9,10]. There are no data that indicate stair descent would have these beneficial effects. At 9.6 METs, stair climbing is a vigorous lifestyle physical activity and a preferable health behavior to descent at half the exercise intensity, 4.7 METs [36]. Two large sample studies have reported effects of signage interventions on descent but not on climbing [26,27]. Combining ascent and descent provides a misleading estimate of the potential public health dividend from these simple interventions [34]. For the studies included in the review by Bellicha and co-workers, increases at work occurred more frequently for stair usage, 81% of the study arms, than stair climbing, 53% of the study arms [3]. Climbing and descent should be analyzed separately to estimate potential effects on population health of signage.

### 1.5. The Current Study

In this paper, aggregated data from automated counters were used to test signage interventions in a number of different workplaces. When a single research group pools data, heterogeneity of methods is less of an issue than it is for any attempt at synthesis across disparate study designs [31]. The interventions were developed from previously successful campaigns for heart health and calorific expenditure [33,35], further refined with focus groups [37]. We predicted that installation of signage would increase stair usage. For the UK data set, minute-by-minute measures of choice were used to estimate effects of momentary pedestrian traffic at the elevator choice-point (*n* = 426,605). We predicted negative effects of momentary traffic on stair climbing [26,29]. In the Barcelona data set, measures of traffic every 30 min were used to estimate effects on behavior of the amount of pedestrian movement within the building (i.e., its ‘busyness’) (*n* = 293,300). We predicted positive effects of ‘busyness’ on stair climbing [26,29]. Measurements taken throughout the working day controlled for a movement pattern characteristic of most buildings of more than one story. Employees arriving for work ascend into the building in the morning and descend during the afternoon on their way out.

The purpose of this study was to investigate the effects of signage on stair use. Analyses addressed two main questions. Effects of signage interventions for stair climbing at work were assessed when the potential influences of pedestrian movement within the building were controlled for in analyses. Secondly, separate analyses of ascent and descent specifically assessed effects on the targeted behavior of stair climbing. We predicted greater effects of signage on stair descent than climbing.

## 2. Methods

The Ethics Subcommittees of the University of Birmingham and Vic-Central University of Catalonia gave ethical approval (ERN_10-1281).

### 2.1. Interventions in the UK

Appendix A summarizes baseline stair usage, structural aspects of the buildings, and the interventions employed for the nine UK workplaces. The employees were predominantly white-collar office staff from six local government and three commercial workforces. Custom built, automatic counters at the ground floor monitored all ascending and descending stair users and all ingress (ascent) and egress (descent) for the elevators. Correlations of the automatic counts.min^−1^ with direct observations were in the range 0.93–0.95 (all *p* < 0.001; see [29]). Only complete data from weekdays were included (7:00–18:59). Far outliers from boxplots of traffic were excluded (up, 0.7%; down, 1.0%).

Baseline and two subsequent campaign phases were of 3-week or 4-week duration. Each phase immediately followed the previous one, with no washout period between them (see Figure 1). The main intervention posters were installed at the choice-point between the stairs and elevators, with a further prompt above the lift button [33]. Campaign content outlined benefits of climbing on heart health, calorific expenditure, or fitness, tested within the period September 2011 to September 2013 (the signage interventions are summarized in Appendix A). 

### 2.2. Interventions in Barcelona, Spain

Appendix A summarizes baseline stair usage, structural aspects of the monitored choice-points and the interventions employed for the three Barcelona workplaces. The employees were predominantly white-collar office and laboratory staff from three commercial workforces. Commercial automatic counters (Solva NL, Amsterdam, The Netherlands) monitored all ascending and descending stair and elevator use, with aggregated data over periods of 30 min provided by Solva. Only complete data from weekdays were included (7:00–19:59). Far outliers from boxplots of traffic were excluded (up, 0.4%; down, 0.6%).

Campaign durations were four weeks, preceded by no-intervention periods (November 2013 to February 2014). The 4-week baseline for the first phase contrasted with a 4-week no-intervention period for the second, two weeks of which occurred before Christmas and two weeks in the following year. The 6-week washout period between campaigns included the Christmas period, for which monitoring was suspended (see flow diagram above). The main intervention posters were installed at the choice-point between stairs and elevators, with further prompts for stair use on the half-landings between floors in the stairwell to enhance any effect [33]. The campaign content outlined benefits of climbing, primarily, on heart health (see Appendix A for the signage interventions).

### 2.3. Data Reduction and Analyses

Pedestrian traffic was mean-centered for each choice-point throughout to avoid confounding traffic with potential effects of other variables. Analyses (2017) employed 3-step hierarchical regression with bootstrapping (1000 samples) to circumvent problems with non-independence of the observations (SPSS version 19 (IBM, Armonk, NY, USA)). In the UK, logistic regressions were used to estimate effects for stair versus elevator choice. On the second step, pedestrian traffic in both directions was added to the first step of intervention alone. The third step added time of day and controlled for structural aspects of the buildings, namely number of elevators (1 vs. 2) and floors (3 to 6). Greater numbers of floors and elevators reduce stair climbing [29,34,38,39]. For Barcelona, multiple regressions were used to estimate effects for the number of stair users. Elevator traffic in both directions on the second step was added to the first step of intervention alone. The third step added time of day and stair use in the opposite direction; similarity of the workplaces precluded analysis of structural aspects.

## 3. Results

### 3.1. Intervention in the UK

Table 1 presents the raw percentage stair use in the different phases, whereas Table 2 summarizes the effects of the first intervention phase relative to baseline with inclusion of the other variables. 

For Table 2, ascent is summarized on the left of the table (up) and descent on the right (down). For stair climbing, a paradoxical reduction in climbing after the intervention was installed (step 1) remained after the inclusion of pedestrian traffic (step 2). In the final model, however, there was no effect of the intervention on climbing (step 3 odds ratio (OR) = 0.99, 95% confidence interval (CI) = 0.97, 1.01). In contrast, the intervention increased stair descent in all three steps, (step 3 OR = 1.10, 95% CI = 1.07, 1.13).

Effects of momentary traffic in the analyses were consistent for both directions of travel. Traffic up was associated with reduced stair use, whereas traffic down was associated with an increase. Effects of time of day were of opposite direction; stair climbing reduced over the working day, whereas stair descent increased. As expected, more elevators and a greater number of floors were associated with reduced stair use. As can be seen from the magnitude of the ORs, the effect of floors was greater for climbing than descent.

Table 3 summarizes the effects of the second intervention phase relative to stair use during the first. In these analyses, a consistent increase in stair climbing contrasted with no intervention effects on stair descent. Effects of momentary traffic, time of day, and structural aspects of the buildings echoed the results during the first intervention phase.

Consistently in these analyses, ascending momentary traffic was associated with reduced stair use. To facilitate comparisons between effects of traffic and the binary intervention variable, traffic was standardized to the range 0–1. The reciprocal of the standardized OR represents increased stair choice at lower levels of traffic. This transformation facilitates comparison of the magnitude of the effects of traffic relative to the magnitude of change produced by signage. For the first intervention phase, a rerun of step 3 provided ORs of 3.390 (95% CI = 3.205, 3.597) and 2.475 (95% CI = 2.458, 2.542) for the effects of ascending traffic on stair climbing and descent respectively. Effects of momentary traffic at the choice-point were at least an order of magnitude greater than the effect of intervention in these standardized analyses for stair climbing (OR = 1.020, 95% CI = 0.998, 1.043) and stair descent (OR = 1.126, 95% CI = 1.099, 1.154). In contrast, descending traffic was associated with an increase in stair use, irrespective of direction. The standardized analyses revealed lower magnitude effects of descending traffic than of ascending traffic on stair use (up OR = 1.214, 95% CI = 1.144, 1.288; down OR = 1.100, 95% CI = 1.038, 1.166).

### 3.2. Intervention in Barcelona

Table 4 summarizes stair use during the different intervention phases, whereas Table 5 and Table 6 summaries analyses that statistically adjusted for effects of traffic (i.e., ‘busyness’) in the first and second intervention phases respectively.

In both intervention phases, an apparent increase in stair climbing after the intervention (step 1) did not survive the addition of elevator traffic (step 2). In contrast, the intervention increased stair descent in all three steps.

In the main, the prediction that increased ‘busyness’ would increase stair use was confirmed. Consistently, elevator traffic in the direction of travel was associated with increased stair climbing and descent, as was stair traffic in the opposite direction. Elevator traffic in the opposite direction, however, was associated with reduced stair climbing and increased stair descent. As in the UK, stair climbing decreased across the day, whereas stair descent increased.

The changes in *R*^2^ revealed substantial increases in explained variance when elevator traffic on the second step was added to the intervention alone. For example, in the first intervention phase, signage alone explained 0.2% of the variance of climbing, whereas elevator traffic explained a further 46.6%. For descent, the equivalent explained variances were 0.3% versus 23.6%.

## 4. Discussion

### 4.1. Effects of Signage Interventions on Stair Use

To summaries the results, we predicted that momentary pedestrian traffic at the choice-point would decrease stair climbing, whereas ‘busyness’ of the building would increase stair climbing, with both effects independent of the signage interventions. Both these predictions about uncontrolled pedestrian movement were confirmed and the hypotheses supported. Increased momentary traffic at the choice-point consistently reduced stair climbing. For ‘busyness’, increased pedestrian movement within the building increased stair usage with one exception; elevator traffic opposite to the direction of travel was associated with reduced stair climbing. These large magnitude effects of pedestrian movement on stair usage were consistent throughout. To avoid distraction from the main question here about effectiveness for public health, discussion of pedestrian movement is presented in Appendix A. Secondly, we predicted that signage would increase stair usage, with effects on stair descent more likely than on climbing. Signage increased descent in three of the four final analyses, whereas effects on climbing only survived the inclusion of the additional variables in one analysis. The hypotheses about the effects of signage were also supported.

This paper pooled workplace studies to estimate effects, as was performed previously for mall-based interventions [31]. Converting the ORs in the earlier study to effects sizes [40] reveals that signage increased climbing in the pooled mall data with a larger effect size, 0.406 (95% CI = 0.325, 0.487) than that of pedestrian traffic over 30 min periods, 0.244 (95% CI = 0.172, 0.315). Uncontrolled pedestrian movement in public access environments where the alternative was the escalator did not preclude large magnitude effects of signage on climbing. In the standardized analyses here of the first UK intervention phase, effect sizes for momentary pedestrian traffic in the direction of travel—up 0.674 (95% CI = 0.643, 0.707), down 0.501 (95% CI = 0.497, 0.515)—were about an order of magnitude greater than the effect sizes for signage alone—up 0.011 (95% CI = 0.001, 0.023), down 0.066 (95% CI = 0.052, 0.079). The results for ‘busyness’ in Barcelona confirm effects of pedestrian movement in the building. The changes in *R*^2^ when ‘busyness’ was added on the second step—up = 0.466, down = 0.236—were orders of magnitude greater than the effects of signage alone—up = 0.002, down = 0.003. Unlike public access sites, pedestrian movement in the workplace has greater effects on stair use than signage. This movement is uncontrollable. Failure to measure pedestrian movement, and control for it in analyses, compromises any estimates of intervention effects. Further, choosing busy periods for assessment at work (e.g., lunchtimes) may have compound the problem in previous research (e.g., [35]), and measurement throughout the working day is required.

The more frequent intervention effects on stair descent than on climbing reinforce empirical indications that stair descent may be an easier intervention target than climbing [26,27]. As noted in the introduction, however, it is only climbing which has demonstrated effects on cardiovascular disease risk [5,6,7,8,9,10]; there is no evidence that descent would produce a similar public health dividend. Stair descent entails more activity than standing in an elevator, but it seems unlikely that employees could accumulate sufficient daily descents to affect health outcomes. Overall, these analyses do not support recommendations for signage interventions at work to enhance population health [11,12]. While the latest review concluded there was strong evidence that ‘prompts to use stairs versus escalators or elevators are effective’ [17] (p. 67), the data here indicate that when the alternative was the elevator, the likely effects on employees were for stair descent, not climbing. This is not an outcome with major health benefits.

Despite this disappointing result, the increases in stair descent, after control for pedestrian movement, offer some encouragement to public health. Point-of-choice prompts for stairs are aids to change that affect those individuals trying to be more active [13,33,38,41,42]. Increased descent suggests some employees were trying to change. Those employees may be better served by individualized workplace interventions (e.g., [9,10]). Stair climbing interventions have had greater effects in the overweight; stairs appear to be a plausible physical activity for those at elevated risk [35,40,43]. This unusual result reflects the fact that most of the population believes they can climb stairs because they already do so as part of daily life. Belief that one can successfully perform a behavior, called self-efficacy, is a major determinant of the behavior [44,45,46,47]. Low self-efficacy, a major deterrent for physical activity, is less likely to be a barrier to stair climbing than it is for formal sport or jogging. At-risk employees could be identified from simple workplace screening by occupational health.

### 4.2. Strengths and Limitations

As noted earlier, any meta-analyses of the effectiveness of stair climbing signage would be hampered by disparate methods. This paper pooled workplace studies to estimate effects, as was performed previously for mall-based interventions [31]. Heterogeneity of methods, recently estimated at *I*^2^ = 97.5% [14], is much less of an issue when a single group pools studies, and commonality of methods is a major strength here. Previously successful intervention messages were refined with employees. Campaign content seems unlikely to explain any intervention failures that were found. Sample sizes with automated counters (up *n* = 368,996; down *n* = 350,909) rivalled those that would be available from previous research for any attempt at meta-analysis (e.g. stair climbing in Bellicha) and co-worker reviews (*n* = 379,491) [3]. There can be no publication bias or file draw for these pooled data. Further, statistical adjustment for pedestrian movement is unique to these new estimates of effect. Failure to produce meaningful increases in climbing in two countries, coupled with large magnitude effects of uncontrolled pedestrian movement, suggests the results will generalize. In the quasi-experimental designs that investigate stair climbing, rival, alternative explanations of change need to be discounted. Uncontrolled effects of pedestrian movement offer a serious rival explanation for changes in stair usage at work, with greater magnitude effects than signage.

The absence of control buildings is a limitation. Nonetheless, consideration of the function of a control group reduces this concern. The only possible external influence on effectiveness, an event coinciding with intervention installation, is implausible; the interventions occurred at three different times over two years in the pooled data set in the UK. Building specific events are implausible, as the interventions occurred simultaneously in at least two different buildings throughout. Concerning potential random influences on the behavior, these analyses include the major random influence, uncontrolled pedestrian movement, for the first time. Effects of uncontrolled movement dwarfed any effects of signage. Any control buildings would also be subject to these uncontrollable effects of pedestrian movement. Given the magnitude of the effects of movement, equating them across conditions with random allocation would be likely to require many more buildings that recruited here. Statistical adjustment for this movement is a more promising approach than a non-intervention control condition without adjustment for the effects of pedestrian movement, which is uncontrollable. These data demonstrate that estimates for the effects of signage at work without statistical adjustment cannot be interpreted with any confidence.

With one exception, measurement of stair choice only occurred at the ground floor. This floor represents the greatest possible height of climb in any building, and an increasing number of floors, (i.e., potential height of climb) was associated with reduced stair climbing, as has been reported previously [34,38,39]. It is possible that increases in behavior occurred for floors on which measurements were not made but where a lower climb would be required. For descent, however, the number of floors had smaller magnitude effects on choice. The increases in stair descent must reflect accumulation of behavior prompted by signage at a number of floors remote from the ground, as increased descent from a single floor would be an infrequent occurrence.

In addition, automated counters cannot measure demographic influences on stair choice such as gender, age, and weight status [48]. Some employees may have responded to the messages, despite the absence of effects overall. Stair interventions may be effective at an individual level, with any effect obscured in the data by larger magnitude influences on the behavior that affects all occupants. Inevitably, interventions only occur when employers accept them, and random sampling of buildings was precluded. Nonetheless, the mixture of local government and commercial office employees makes any bias less likely. Finally, neither data set could be used to accurately estimate the influence of floors or elevators. The variables were inextricably confounded in the UK; taller buildings that accommodate more occupants require more elevators. The Barcelona buildings lacked variance. Only stratified sampling for differences in the number of elevators, but matches on the number of floors, could accurately test effects of structural features of buildings.

## 5. Conclusions

Stair climbing interventions at work with signage are promoted as a simple tool to increase lifestyle physical activity. Stair climbing is a very cost-effective intervention [49], despite reservations about its potential population impact [49,50]. Only the repeated episodes of stair climbing that are available at work could provide meaningful public health dividends from signage interventions [18]. In these data, small magnitude changes at work occurred, with increased stair descent more frequently found than stair climbing. These pooled estimates provide limited encouragement for the use of signage interventions at the point-of-choice in the workplace to improve health at a population level.

## Figures and Tables

**Figure 1 ijerph-16-03782-f001:**
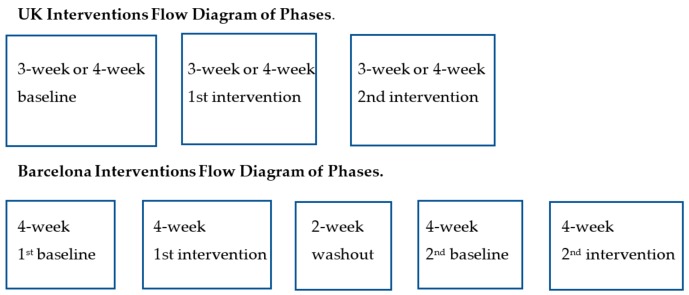
Flow diagrams of the intervention phases in the UK and Barcelona.

**Table 1 ijerph-16-03782-t001:** Percentage stair use (95% CIs) for the different intervention phases in the UK.

Behavior	Baseline	Intervention 1	Intervention 2
Stair climbing (%)	45.1 ^a^ (44.7, 45.5)	43.3 (42.8, 43.7)	44.8 (44.4, 45.2)
Stair descending (%)	67.7 (67.4, 68.1)	69.8 (69.4, 70.2)	70.1 (69.7, 70.5)

^a^ These are raw percentages, uncorrected for the effects of traffic, time of day, or structural aspects of the buildings.

**Table 2 ijerph-16-03782-t002:** Summary of the effects of variables on stair use in the UK during the first intervention phase.

Variable	Up Step 1	Up Step 2	Up Step 3	Down Step 1	Down Step 2	Down Step 3
1st intervention (baseline)	0.926 ^a^ *** (0.906, 0.946)	0.919 *** (0.899, 0.939)	0.989 (0.968, 1.011)	1.101 *** (1.075, 1.129)	1.098 *** (1.070, 1.124)	1.098 *** (1.069, 1.125)
Traffic up (min^−1^)	-	0.911 *** (0.906, 0.915)	0.902 *** (0.898, 0.908)	-	0.934 *** (0.929, 0.940)	0.930 *** (0.924, 0.937)
Traffic down (min^−1^)	-	1.027 *** (1.020, 1.033)	1.042 *** (1.035, 1.049)	-	1.031 *** (1.023, 1.037)	1.029 *** (1.022, 1.035)
Time of day (hour)	-	-	0.974 *** (0.970, 0.978)	-	-	1.010 *** (1.006, 1.015)
Number of elevators	-	-	0.659 *** (0.640, 0.677)	-	-	0.655 *** (0.633, 0.677)
Number of floors	-	-	0.761 *** (0.754, 0.770)	-	-	0.952 *** (0.942, 0.963)

^a^ Odds ratios (95% CIs) and significance levels from bootstrap estimates are shown. ***** = *p* < 0.05, ****** = *p* < 0.01, and ******* = *p* ≤ 0.001.

**Table 3 ijerph-16-03782-t003:** Summary of the effects of variables on stair use in the UK during the second intervention phase.

Variable	Up Step 1	Up Step 2	Up Step 3	Down Step 1	Down Step 2	Down Step 3
2nd intervention (1st intervention)	1.064 ^a^ *** (1.042, 1.088)	1.082 *** (1.059, 1.104)	1.059 ** (1.037, 1.082)	1.017 (0.993, 1.041)	1.019 (0.996, 1.044)	1.014 (0.991, 1.039)
Traffic up (min^−1^)	-	0.886 *** (0.881, 0.890)	0.876 *** (0.871, 0.881)	-	0.911 *** (0.905, 0.916)	0.908 *** (0.902, 0.913)
Traffic down (min^−1^)	-	1.009 ** (1.003, 1.016)	1.031 *** (1.023, 1.039)	-	1.016 *** (1.009, 1.022)	1.014 *** (1.007, 1.020)
Time of day (hour)	-	-	0.975 *** (0.970, 0.978)	-	-	1.011 *** (1.006, 1.015)
Number of elevators	-	-	0.694 *** (0.675, 0.714)	-	-	0.689 *** (0.668, 0.712)
Number of floors	-	-	0.751 *** (0.744, 0.758)	-	-	0.978 *** (0.968, 0.989)

^a^ Odds ratios (95% CIs) and significance levels from bootstrap estimates are shown. ***** = *p* < 0.05, ****** = *p* < 0.01, and ******* = *p* ≤ 0.001.

**Table 4 ijerph-16-03782-t004:** Mean stair use (95% CIs) every 30 min for the different intervention stages in Barcelona.

Behavior	Baseline 1	Intervention 1	Washout Period	Intervention 2
Stair climbing.30 min^−1^	7.22 ^a^ (6.82, 7.61)	8.09 (7.79, 8.39)	8.56 (8.17, 8.94)	8.70 (8.38, 9.03)
Stair descent.30 min^−1^	10.37 (9.95, 10.81)	11.11 (10.81, 11.42)	10.80 (10.43, 11.17)	11.16 (10.83, 11.48)

^a^ These are raw means, uncorrected for the effects of traffic or time of day.

**Table 5 ijerph-16-03782-t005:** Summary of the effects of variables on stair use in Barcelona during the first intervention phase.

Variable	Up Step 1	Up Step 2	Up Step 3	Down Step 1	Down Step 2	Down Step 3
1st intervention (baseline)	0.092 ** ^a^ (0.026, 0.164)	0.047 (−0.003, 0.098)	0.040 (−0.009, 0.086)	0.118 *** (0.048, 0.183)	0.069 * (0.005, 0.131)	0.058 * (0.002, 0.115)
Elevator traffic in direction of travel (30 min^−1^)	-	0.723 *** (0.689, 0.758)	0.608 *** (0.574, 0.643)	-	0.483 *** (0.451, 0.517)	0.423 *** (0.384, 0.462)
Elevator traffic opposite to direction of travel (30 min^−1^)	-	−0.115 *** (−0.143, −0.089)	−0.100 *** (−0.129, −0.071)	-	0.062 ** (0.029, 0.094)	−0.000 (−0.048, 0.045)
Stair traffic opposite to direction of travel (30 min^−1^)	-	-	0.130 *** (0.098, 0.157)	-	-	0.198 *** (0.152, 0.241)
Time of day (hour)	-	-	−0.253 *** (−0.275, −0.230)	-	-	0.192 *** (0.163, 0.221)
Change in *R*^2^	0.002 **	0.466 ***	0.058 ***	0.003 **	0.236 ***	0.037 ***

^a^ Standardised coefficients (95% CIs) and significance levels from bootstrap estimates are shown. * = *p* < 0.05, ** = *p* < 0.01, *** = *p* ≤ 0.001.

**Table 6 ijerph-16-03782-t006:** Summary of the effects of variables on stair use in Barcelona during the second intervention phase.

Variable	Up Step 1	Up Step 2	Up Step 3	Down Step 1	Down Step 2	Down Step 3
2nd intervention (washout).	0.077 ** ^a^ (0.016, 0.141)	0.021 (−0.026, 0.065)	0.019 (−0.025, 0.063)	0.114 *** (0.053, 0.175)	0.065 * (0.018, 0.114)	0.055 * (0.008, 0.101)
Elevator traffic in direction of travel (30 min^−1^)	-	0.632 *** (0.598, 0.665)	0.531 *** (0.501, 0.565)	-	0.491 *** (0.461, 0.522)	0.462 *** (0.432, 0.494)
Elevator traffic opposite to direction of travel (30 min^−1^)	-	−0.033 * (−0.059, −0.005)	−0.058 ** (−0.089, −0.028)	-	0.130 *** (0.103, 0.158)	0.073 *** (0.035, 0.106)
Stair traffic opposite to direction of travel (30 min^−1^)	-	-	0.149 *** (0.121, 0.179)	-	-	0.184 *** (0.152, 0.218)
Time of day (hour)	-	-	−0.265 *** (−0.286, −0.244)	-	-	0.192 *** (0.169, 0.215)
Change in *R*^2^	0.001 **	0.415 ***	0.070 ***	0.003 **	0.340 ***	0.037 ***

^a^ Standardised coefficients (95% CIs) and significance levels from bootstrap estimates are shown. * = *p* < 0.05, ** = *p* < 0.01, and *** = *p* ≤ 0.001.

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
