# Peer review of "Signage Interventions for Stair Climbing at Work: More than 700,000 Reasons for Caution"

_ijerph, 2019, doi:10.3390/ijerph16193782_

Round 1

Reviewer 1 Report

I really enjoyed your manuscript.  Good work!  A suggestion for you is a the end of your introduction/background would to more simply add the phrase " the purpose of this study was to investigate the effects of signage on climbing stairs".  Something like that.  I feel that this simple statement helps frame the reader what the study is about.  

Also, you should add a study or two from the pedometer literature regarding number of steps taken in activity.

Overall, good work!

Author Response

We thank the reviewer for their kind words. 

We have added the sentence 'The purpose of this study was to investigate the effects of signage on stair use.' at the start of the final paragraph before the methods section.

Although the reviewer requested that we insert some information about pedometer studies, we have not done so.  The paper is about stair usage which usually makes a minor contribution to step counts.  We are unsure how information about pedometers would enhance the paper.  

Reviewer 2 Report

Puig-Ribera and colleagues addressed two questions: 1) Effects of signage interventions for stair climbing at work and 2) Separate analyses of ascent and descent specifically assessed effects on the targeted behavior of stair climbing. The manuscript is well written. I have no major issues. Please see below for specific comments.

Introduction

Line 55: (p.67) should be deleted.

Line 58: Discard (see also below)

Lines 57-60: This paragraph should be cited.

Methods

Details of the sample should be addressed.

A diagram of the interventions would help to the comprehension of the reader.

Results

The columns of the tables 5 and 6 should be well justified and zeros should appear before decimal numbers as is detailed in Tables 2 and 3, respectively.

Discussion

Well written

Author Response

We thanks the reviewer for their comments which have improved the clarity of the manuscript

Line 55: (p.67) should be deleted.  We have retained the information about the page number as this referred to a quotation

Line 58: Discard (see also below).  As requested, we have discarded the phrase in parentheses

Lines 57-60: This paragraph should be cited. We have added the citations as requested.

We have expanded the information on the sample as requested, e.g. 'The employees were predominantly white-collar, office staff from six local government and three commercial workforces.'

We have added a flow diagram of the intervention phases in the UK and Barcelona as requested.

We have added the missing zeros to tables 5 and 6 as requested.